

# Effect of major ozone autohemotherapy in fibromyalgia syndrome: a retrospective study

Oktay Faysal Tertemiz[1], Erkan Ozduran[2], Sinan Gursoy[3], Volkan Hanci[4], Kenan Kaygusuz[5], Ilhan Celil Ozbek[6], Mustafa Kurçaloğlu[7] and Ülkü Sabuncu[8]

[1] Anesthesiology and Reanimation, Pain Medicine, Izmir City Hospital, Izmir, Turkey
[2] Physical Medicine and Rehabilitation, Pain Medicine, Sivas Numune Hospital, Sivas, Turkey
[3] Anesthesiology and Reanimation, Pain Medicine, Cumhuriyet University, Sivas, Turkey
[4] Anesthesiology and Reanimation, Critical Care Medicine, Dokuz Eylül University, Izmir, Turkey
[5] Anesthesiology and Reanimation, Critical Care Medicine, Cumhuriyet University, Sivas, Turkey
[6] Physical Medicine and Rehabilitation, Health Science University, Derince Training and Research Hospital, Kocaeli, Turkey
[7] Anesthesiology and Reanimation, Pain Medicine, Ondokuz Mayis University, Samsun, Turkey
[8] Anesthesiology and Reanimation, Pain Medicine, Ankara Bilkent City Hospital, Ankara, Turkey

Corresponding author
Erkan Ozduran,
erkanozduran@gmail.com

## ABSTRACT

**Background:** Fibromyalgia Syndrome (FMS) is a multifaceted chronic disease characterized by widespread musculoskeletal pain, fatigue, sleep disturbances, cognitive dysfunction, and increased sensitivity to stimuli, commonly referred to as central sensitization. Ozone therapy is currently used in the treatment of many chronic diseases by creating controlled oxidative stress. This study aimed to evaluate the effects of major autohemotherapy (MAH) on pain, quality of life, functional status, and sleep quality in patients with FMS.

**Methods:** A total of 72 patients who underwent 10 sessions of MAH between April 1, 2023 and September 1, 2023 were included in this retrospective study. A total of 10 sessions of major ozone therapy were applied to the patients, two sessions per week. In the first session, a total of 1,000 mcg of MAH (100 ml at 10 mcg/ml) was administered. In the following sessions, the concentration was gradually increased by 5 mg/ml at each session reaching up to 30 mcg/ml with a maximum cumulative dose of 3,000 mcg. Pre-and post-treatment pain status of the patients was evaluated with the Visual Analog Scale (VAS), functional status with the Fibromyalgia Impact Questionnaire (FIQ), quality of life with the Short Form-36 (SF-36), and sleep status with the Pittsburgh Sleep Quality Index (PSQI).

**Results:** The mean age of 68 female and 4 male patients included in the study was determined as 54.17 ± 11.17. VAS was determined as 7.76 ± 2.01 before treatment and 4.77 ± 2.20 after treatment. FIQ score was determined as 69.08 ± 16.01 before treatment and 56.18 ± 22.46 after treatment, Pittsburg Sleep Quality Index score was determine as 10.27 ± 3.37 before treatment and 9.61 ± 2.81 after treatment. Statistically significant differences were found in VAS, FIQ, SF-36, and PSQI values before and after treatment ($p < 0.001$).

**Conclusion:** According to the study results, MAH has been found to provide significantly positive effects in reducing pain intensity, improving functional status, enhancing quality of life, and improving sleep patterns in individuals diagnosed with

fibromyalgia syndrome. Considering the low side effect profile and positive clinical results of MAH in fibromyalgia patients resistant to other treatment methods, it is recommended that this method be considered as a complementary or alternative treatment option.

# INTRODUCTION

Fibromyalgia Syndrome (FMS) is a multifaceted chronic disease characterized by widespread musculoskeletal pain, tenderness in specific anatomical regions, fatigue, sleep disturbances, cognitive dysfunction, and increased sensitivity to stimuli, often referred to as central sensitization. A combination of genetic predisposition, environmental factors, and neurophysiological abnormalities contribute to the onset and progression of the disease (*Wasti et al., 2023*; *Dizner-Golab, Lisowska & Kosson, 2023*). Furthermore, FMS imposes a significant healthcare burden. A large US database analysis found that 12-month average total healthcare costs were approximately three times higher in patients with FMS compared to those without FMS (*Lloyd et al., 2012*). In the United States, standardized estimates of the average annual cost per patient diagnosed with fibromyalgia ranged from $1,750 to $35,920 when restricted to healthcare expenditures, whereas analyses incorporating overall total direct costs reported values between $6,208 and $9,505 (*D'Onghia et al., 2022*). Ozone therapy is growing in popularity due to its effectiveness, ease of application, low cost, and few significant side effects (*Hidalgo-Tallón et al., 2022*).

FMS treatment is divided into two categories: pharmacological and non-pharmacological agents. Pharmacological agents include tricyclic antidepressants such as amitriptyline and nortriptyline, selective serotonin and norepinephrine uptake inhibitors (SNRIs) such as duloxetine, anticonvulsant drugs such as pregabalin, and, in some cases, opioid drugs such as tramadol. Non-pharmacological agents include cognitive-behavioral therapy, mindfulness therapy, exercise therapy such as aerobic training, tai chi, yoga, qigong, acupuncture, and hypnosis (*Filipovic et al., 2025*).

Major autohemotherapy (MAH) has emerged as a new therapeutic option in recent years due to its potential systemic effects along with other ozone treatments. MAH activates the immune system, supports cellular repair processes, regulates oxygen metabolism, and reduces oxidative stress (*Ahi & Afsar, 2022*; *Saime, Tur & Evcik, 2019*). Ozone therapy has become more popular in recent years due to its ease of application and low side effect profile (*Deepthi & Bilichodmath, 2020*; *Hidalgo-Tallón et al., 2022*).

Ozone therapy has been reported to have positive short- and medium-term effects on pain, fatigue, and sleep disturbances in fibromyalgia patients. Its antioxidant, anti-inflammatory, and immunomodulatory properties are reported to be effective in the pathogenesis of fibromyalgia (*Üşen et al., 2025*). Ozone therapy can be administered not only *via* intravenous routes but also through subcutaneous, intradiscal, intra-articular, and intramuscular applications to effectively target specific tissues. In addition, urethral,

vaginal, and rectal insufflation, as well as both major and minor autohemotherapy protocols, represent other commonly employed approaches in clinical practice (*Bocci, 2006*). Researchers suggest that medical ozone injected into the musculoskeletal system for pain treatment inhibits pain signaling by oxidizing algogenic receptors and activating the antinociceptive system, underlying its anti-inflammatory, analgesic, and antiedema properties. There are no standard dosage protocols. In injection applications, gas concentrations generally vary between 4 and 30 μg/ml, and amounts vary between 5 and 15 ml. The number of infiltration sessions typically varies between 3 and 10 (usually 1 or 2 per week), depending on the specific circumstances of each case. This lack of standardization in treatment protocols makes it difficult to compare results during a systematic review and prevents high-quality conclusions or recommendations. Almost all researchers agree on the high safety of ozone therapy, especially when performed with modern medical ozone generators (*Hidalgo-Tallón et al., 2022*).

In myofascial pain syndrome, ozone injection into painful muscles, along with steroid, dry needling, or steroid application, are currently popular research topics. Application of ozone and/or steroid to myofascial bands has been reported to provide significant improvements in Visual Analog Scale (VAS) scores. However, it has been reported in the literature that in the third-week VAS results of combined ozone and steroid injections, no difference was observed compared to the ozone-only treatment. This may be due to the delayed effect of steroids, meaning their anti-inflammatory effects take longer to manifest (*Eldemrdash et al., 2024*). Ozone, lidocaine, and dry needling treatments applied to myofascial trigger points in the neck region were all found to be highly effective in improving patient pain and pain pressure threshold (PPT). A slightly greater improvement in VAS, PPT, and Neck Disability Index was found in the ozone therapy group compared to the lidocaine group, without statistically significant differences (*Jandura et al., 2024*). Repeated intra-articular ozone injections have been shown to lead to improvements in pain, function, and range of motion in shoulder adhesive capsulitis, which were not statistically different from a single corticosteroid injection (*Sahillioğlu, Ayyıldız & Şahin, 2025*). All studies emphasize the need for future large, multicenter, randomized trials with standardized protocols to clarify the comparative results of these methods and confirm their effectiveness.

The antioxidant and immunomodulatory effects of ozone therapy may target the underlying mechanisms of FMS, providing relief from many clinical conditions, such as pain relief and functional recovery (*Ahi & Afsar, 2022*; *Dıraçoğlu, 2016*; *Sucuoğlu & Soydaş, 2023*). However, research evaluating the effectiveness of ozone therapy in the treatment of FMS is limited. While existing studies offer promising results, more comprehensive and high-quality studies are needed to determine the benefits of the treatment.

The aim of this study was to retrospectively evaluate the effectiveness of MAH on pain, functional capacity, sleep, and quality of life in patients diagnosed with medically refractory FMS and to further investigate the potential impact of MAH. This study aims to contribute to the scientific knowledge of ozone therapy's place in modern treatment approaches by examining its potential to alleviate fibromyalgia symptoms.

## METHODS

This study was planned in accordance with the Declaration of Helsinki, in line with the approval number 2023/09-11 GOA (Date: 21.09.2023) received from Sivas Cumhuriyet University Non-Interventional Ethics Committee. The records of patients diagnosed with fibromyalgia who were resistant to medical treatments and received MAH treatment, who applied to Sivas Numune Hospital Traditional and Complementary Medicine polyclinic and Algology polyclinic between 1 April 2023 and 01 September 2023, were analyzed retrospectively.

Patients diagnosed with FMS according to the American College of Rheumatology (ACR) 2016 revised diagnostic criteria were accepted to the study. Inclusion criteria included being between the ages of 18 and 80, being unresponsive to medical treatment for at least 3 years (drug treatments: antidepressants, antiepileptics, non-drug treatments: acupuncture, aerobic activity, cognitive-behavioral therapy), being literate, and accepting treatment. Exclusion criteria are; patients who have glucose-6-phosphate dehydrogenase deficiency, have hemoglobin level lower than 7 g/dl, have high thyroid function, are illiterate, have a history of allergy, are pregnant or breastfeeding women, use anticoagulants and do not accept MAH treatment. Participants who did not receive other treatments while receiving ozone therapy were included in the study due to concerns that the results of ozone therapy treatment might be affected if applied together with other fibromyalgia treatments. MAH was performed by a physician who has an ozone application certificate. Our study was designed as a double-blind study, and neither the participants nor the administering physician were aware of the ozone concentration. Dose adjustments and injection preparation were performed by an independent physician, and only this physician knew the dose to be administered.

### Study design

Written and verbal informed consent was obtained from each patient before MAH. Demographic data of the patients, such as age and gender, were recorded. All surveys were administered face-to-face twice, one day before the first MAH treatment and three days after the completion of 10 sessions of MAH treatment.

Patients who met the inclusion criteria were evaluated in terms of demographic data (gender, age). Then, the patients were evaluated in terms of pain, functional capacity, sleep status, and quality of life. VAS was used to assess pain intensity, FIQ to assess functional capacity, Pittsburgh Sleep Quality Index (PSQI) to assess sleep quality, and Short Form-36 (SF-36) to assess quality of life. FIQ, PSQI, SF-36 and VAS scores were recorded once before and after the treatment.

Patients who met the inclusion criteria were evaluated for demographic data (gender and age). The patients were then evaluated in terms of pain, functional capacity, sleep status and quality of life. VAS was used to assess pain intensity, FIQ to assess functional capacity, Pittsburgh Sleep Quality Index (PSQI) to assess sleep quality, and Short Form-36 (SF-36) to assess quality of life. FIQ, PSQI, SF-36 and VAS scores were recorded once before and after the treatment (*Totten & Wondzi, 2025*; *Tirelli et al., 2019*).

## Evaluation criteria and surveys

**Functional status:** The Turkish version of the FIQ was used to assess functional status. This questionnaire, used to assess disease course, response to treatment, and, more importantly, disease impact in patients with fibromyalgia, has a maximum score of 100. The 10 different characteristics assessed on this scale are the number of days patients feel well, physical function, number of days off work, pain, fatigue, morning sickness, stiffness, depression, and anxiety. Higher scores on this scale and lower scores on other parameters have been associated with better functional status (*Ahi & Afsar, 2022*; *Sucuoğlu & Soydaş, 2023*).

**Sleep:** The Pittsburgh Sleep Quality Index (PSQI) was used to assess sleep status. The questionnaire, which assesses sleep quality over the past month, questions components such as subjective sleep quality, sleep duration, use of sleeping medication, sleep disturbances, sleep latency, daytime dysfunction, and habitual sleep efficiency. Each component is scored between 0 and 3, and the final PSQI score is between 0 and 21. A total PSQI score of 5 points or more and high scores in the above-mentioned components are associated with poor sleep quality (*Sucuoğlu & Soydaş, 2023*).

**Quality of life:** The Short Form-36 (SF-36) was used to assess health-related quality of life. The questionnaire, which has been validated and validated in Turkish, measures nine parameters, with higher scores indicating better quality of life. These nine parameters can be listed as follows: physical functioning (PF), vitality (VT), health change (HC), social functioning (SF), role physical (RP), mental health (MH), general health (GH), role emotional (RE), and pain (P) (*Ahi & Afsar, 2022*; *Gazioğlu Türkyılmaz, Rumeli & Bakır, 2021*).

## Application methods

Ozone dose, concentration and number of sessions in MAH interventions were applied in accordance with the recommendations of the Madrid Declaration on Ozone Therapy (MDOT) (*International Scientific Committee of Ozone Therapy, 2015*). A total of 10 sessions of major ozone therapy were applied to the patients, two sessions per week. In the first session, a total of 1,000 mcg of MAH (100 ml at 10 mcg/ml) was administered. In the following sessions, the concentration was gradually increased by 5 mg/ml at each session reaching up to 30 mcg/ml with a maximum cumulative dose of 3,000 mcg (*Gazioğlu Türkyılmaz, Rumeli & Bakır, 2021*).

In our study, the Turkozone Blue S ozone generator, approved by the Ministry of Health of the Republic of Turkey, was used (*Sucuoğlu & Soydaş, 2023*). Before MAH was applied, 100 ml of autologous blood was filled from the patient's vein in the antecubital region into a vacuum-containing glass bottle suitable for ozone therapy containing 10 ml of citrate. 100 ml of ozone gas at the appropriate concentration for the session was drawn into the ozone injector and injected into the bottle filled with patient blood. After the ozonization process of the blood was completed, the infusion was completed within 15 min through the same vein to the patient with a suitable blood filter tube (*Sucuoğlu & Soydaş, 2023*).

## Statistical analysis

The data of our study were analyzed using the Statistical Package for Social Sciences (SPSS) version 29.0 statistical package program (IBM Corp., Armonk, NY, USA). Values taking continuous data are mean ± standard deviation; median (minimum–maximum) and frequency data are expressed as number ($n$) and percentage (%). The distribution pattern of the data taking continuous values was determined by Kolmogorov Smirnov and Shapiro Wilk tests. According to the distribution pattern, Student's t test and Mann Whitney U test were used in the intergroup analysis of the data with continuous values, and paired sample T-test and Wilcoxon Signed Rank Test were used in the intragroup analysis. A $p$ value of less than 0.05 was considered statistical significance.

## Power analysis

The number of patients to be included in our study was determined with power analysis. For power analysis, the website https://clincalc.com/stats/samplesize.aspx and the power analysis program found there were used. The hypothesis of our study was determined as ozone therapy reducing the FIQ score. The main outcome of our study was determined as the FIQ score after ozone treatment. A study by *Gazioğlu Türkyılmaz, Rumeli & Bakır (2021)* determined the FIQ score before ozone treatment in fibromyalgia patients as 62.95 ± 10.40. As determined by *Gazioğlu Türkyılmaz, Rumeli & Bakır (2021)* for the FIQ score before ozone treatment was 62.95 ± 10.40, value with alpha error 0.05, 7% difference between the ozone treatment and 95% power, required a sample number of at least 72 individuals for the group.

## RESULTS

Of the 81 patients included in the study, six could not complete the study due to their personal reasons, and three of them could not complete the study due to feeling unwell during MAH, allergic reactions, and headaches after treatment (Fig. 1). Statistical analysis was performed by applying 10 sessions of MAH to 72 patients included in the study. Of the patients included in our study, 68 (94.4%) were women and four (5.6%) were men. The average age of all participants was determined as 54.17 ± 11.17.

In the pain score evaluation, the mean VAS before treatment was 7.76 ± 2.01, while it was 4.77 ± 2.20 after treatment. A statistically significant difference was found between the VAS values before and after treatment ($p < 0.001$) (Fig. 2).

In the functional status assessment, the mean FIQ score was determined as 69.08 ± 16.01 before treatment and 56.18 ± 22.46 after treatment. A significant difference was found between the pre-treatment FIQ scores and the post-treatment FIQ scores ($p < 0.001$) (Fig. 3).

A significant difference was found between pre- and post-treatment values in all parameters of the SF-36 questionnaire used in quality of life assessment ($p < 0.001$) (Fig. 4).

The pre-treatment average of the Pittsburg Sleep Quality Index used in sleep status evaluation was 10.27 ± 3.37 and the post-treatment average was 9.61 ± 2.81. Similarly, a significant difference was found in the pre-treatment and post-treatment sleep questionnaire results ($p < 0.001$) (Fig. 5).

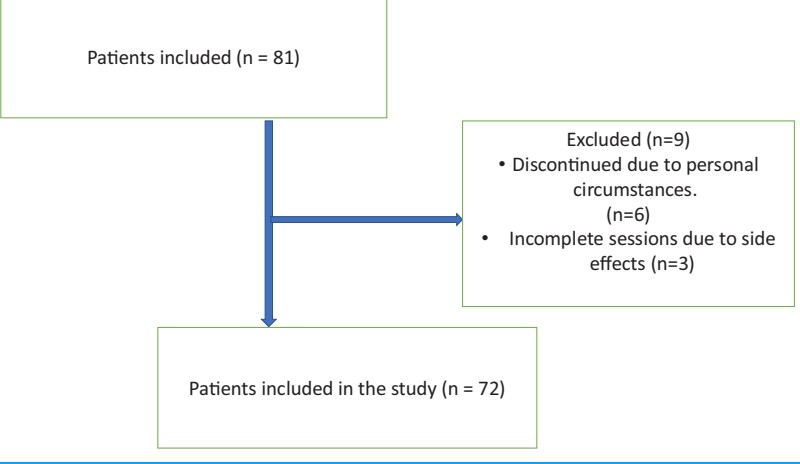

**Figure 1  Flow chart of participants.**               

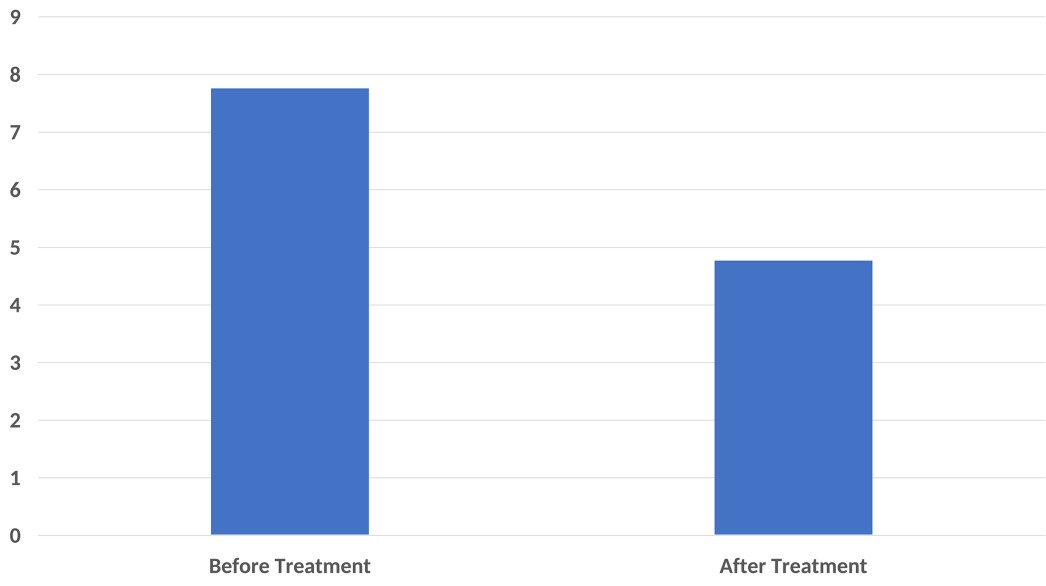

**Figure 2  A significant decrease in pain score compared to base-line after ten sessions of major autohemotherapy application, *P* < 0.001.**      

The mean age of patients included in our study was 49.91 ± 7.7 years for those under 65 years of age, and 71.21 ± 4.56 years for those 65 and over. No significant difference was found in VAS, FIQ, and SF-36 (except Physical Functioning) scores between the groups under and over 65 years of age before and after treatment ($p > 0.005$). Pre- and post-treatment PSQI scores showed significant differences between the groups under and over 65 years of age ($p = 0.012$ and $p = 0.013$, respectively).

In the evaluation made with Pearson correlation analysis, a positive moderate correlation relationship was found between the Pittsburgh Sleep Quality Index and the pre- and post-treatment values of the energy/liveliness/vitality and mental health sub-parameters of SF-36 (Table 1).

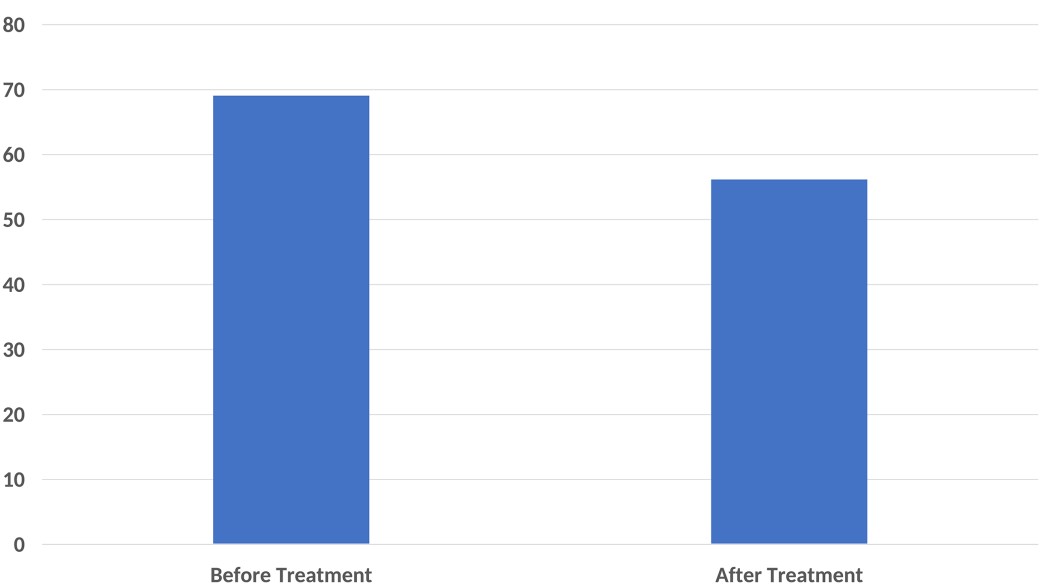

**Figure 3 A significant decrease in Fibromyalgia Impact Questionnaire (FIQ) score was found compared to the baseline after ten sessions of major autohemotherapy application, *P* < 0.001.**

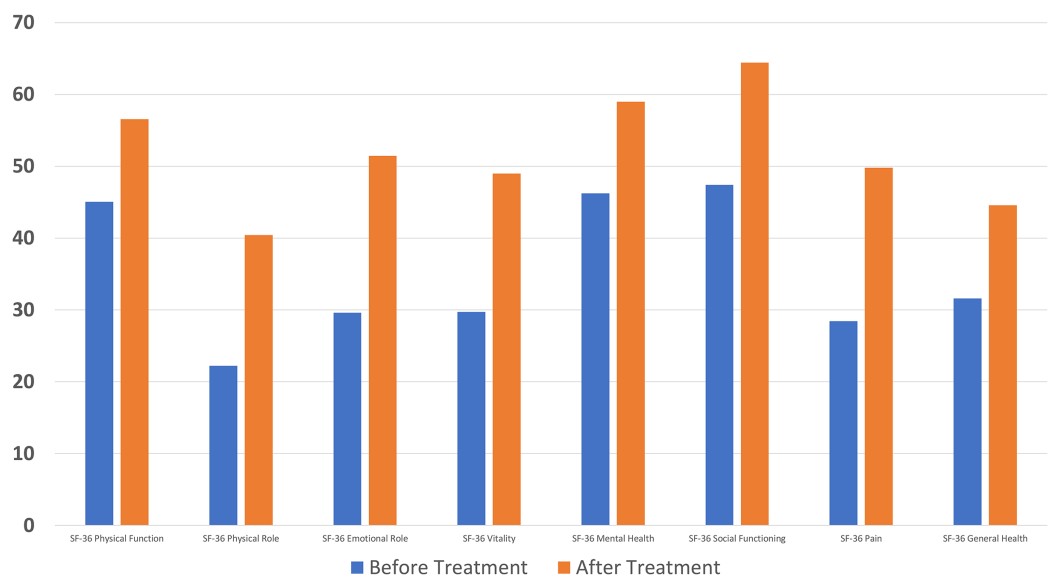

**Figure 4 A significant decrease in Short Form-36 score was found compared to the baseline after ten sessions of major autohemotherapy application, *P* < 0.001.**

In the evaluation made with Spearmen correlation analysis, a positive weak correlation relationship was found between the pre- and post-treatment values in the physical role difficulty, emotional role difficulty and pain sub-parameters of VAS and SF-36 (Table 2).

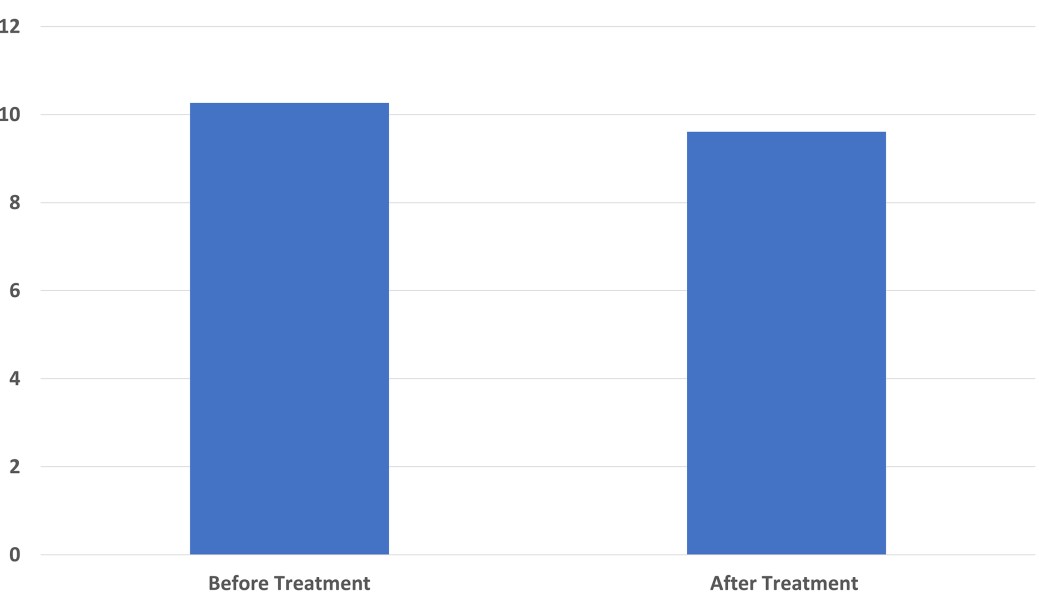

**Figure 5** A significant decrease in Pittsburgh Sleep Quality Index score score was found compared to the baseline after ten sessions of major autohemotherapy application, *P* < 0.001.

## DISCUSSION

Patients with FMS have a low quality of life due to pain, sleep disorders, and depression. In this study, we evaluated the efficacy of MAH in patients with FMS and achieved encouraging clinical results. According to the study results, we found that MAH has significant positive effects on pain, functional status, quality of life, and sleep in the treatment of FMS. MAH appears to be an alternative treatment option for patients with FMS due to its ease of application and low probability of serious side effects or complications.

FMS is defined as a central sensitivity syndrome and is clinically manifested by the presence of chronic pain, sleep disorders, anxiety, fatigue and depressive episodes (*Maffei, 2020*). The pathogenesis of FMS is also complex and both genetic and environmental factors seem to play a role in the pathophysiology of the disease (*Ablin, Neumann & Buskila, 2008*). There is evidence that oxidative stress is increased in FMS, but the cause of this increase is unknown (*Ozgocmen et al., 2006*). Imbalance between oxidative and antioxidative processes, increased prooxidative stress, increased $H_2O_2$ and lipid peroxidation products play a role in the pathogenesis of fibromyalgia (*Assavarittirong, Samborski & Grygiel-Górniak, 2022*). Ozone therapy upregulates endogenous antioxidative systems, suppresses inflammatory processes and strengthens the immune system through the controlled oxidative stress it provides (*Tirelli et al., 2019*). Oxidative stress and nitric oxide play an important role in the pathophysiology of fibromyalgia (*Ozgocmen et al., 2006*). According to these mechanisms of action, it has been hypothesized that ozone therapy may be beneficial in the management of fibromyalgia (*Sucuoğlu & Soydaş, 2023*;

**Table 1 Pearson correlation analysis between normally distributed data.**

| | | Age | Pittsburg Sleep Quality Index-BT | Pittsburg Sleep Quality Index-AT | Fibromyalgia Impact Questionnaire-BT | Fibromyalgia Impact Questionnaire-AT | SF-36-Energy-BT | SF-36-Energy-AT | SF-36-Mental Health-BT | SF-36-Mental Health-AT |
|---|---|---|---|---|---|---|---|---|---|---|
| Age | r | 1 | **−0.254*** | −0.189 | −0.101 | −0.033 | −0.011 | −0.122 | −0.004 | −0.117 |
| | p | | **0.036** | 0.132 | 0.407 | 0.788 | 0.926 | 0.312 | 0.971 | 0.335 |
| Pittsburg Sleep Quality Index-BT | r | **−0.254*** | 1 | **0.674**** | 0.054 | 0.064 | **−0.381**** | −0.066 | **−0.425**** | −0.229 |
| | p | **0.036** | | **0.000** | 0.660 | 0.604 | **0.001** | 0.593 | **0.000** | 0.060 |
| Pittsburg Sleep Quality Index-AT | r | −0.189 | **0.674**** | 1 | −0.036 | 0.030 | −0.238 | −0.198 | **−0.268*** | −0.228 |
| | p | 0.132 | **0.000** | | 0.779 | 0.811 | 0.056 | 0.114 | **0.031** | 0.067 |
| Fibromyalgia Impact Questionnaire-BT | r | −0.101 | 0.054 | −0.036 | 1 | 0.148 | −0.142 | −0.087 | **−0.251*** | −0.185 |
| | p | 0.407 | 0.660 | 0.779 | | 0.223 | 0.233 | 0.475 | **0.033** | 0.125 |
| Fibromyalgia Impact Questionnaire-AT | r | −0.033 | 0.064 | 0.030 | 0.148 | 1 | −0.032 | 0.019 | −0.114 | −0.032 |
| | p | 0.788 | 0.604 | 0.811 | 0.223 | | 0.795 | 0.877 | 0.348 | 0.790 |
| SF-36-Energy-BT | r | −0.011 | **−0.381**** | −0.238 | −0.142 | −0.032 | 1 | **0.508**** | **0.660**** | **0.448**** |
| | p | 0.926 | **0.001** | 0.056 | 0.233 | 0.795 | | **0.000** | **0.000** | **0.000** |
| SF-36-Energy-AT | r | −0.122 | −0.066 | −0.198 | −0.087 | 0.019 | **0.508**** | 1 | **0.369**** | **0.588**** |
| | p | 0.312 | 0.593 | 0.114 | 0.475 | 0.877 | **0.000** | | **0.002** | **0.000** |
| SF-36-Mental Health-BT | r | −0.004 | **−0.425**** | **−0.268*** | **−0.251*** | −0.114 | **0.660**** | **0.369**** | 1 | **0.612**** |
| | p | 0.971 | **0.000** | **0.031** | **0.033** | 0.348 | **0.000** | **0.002** | | **0.000** |
| SF-36-Mental Health-AT | r | −0.117 | −0.229 | −0.228 | −0.185 | −0.032 | **0.448**** | **0.588**** | **0.612**** | 1 |
| | p | 0.335 | 0.060 | 0.067 | 0.125 | 0.790 | **0.000** | **0.000** | **0.000** | |

Note:
BT: Before Treatment, AT: After Treatment, bold font: statistically significant, *: correlation is significant below 0.05, **: correlation is significant below 0.001.

*International Scientific Committee of Ozone Therapy, 2015*; *Tirelli et al., 2019*; *Gazioğlu Türkyılmaz, Rumeli & Bakır, 2021*). The beneficial effects observed with ozone therapy were surprising, showing clinically significant improvements in the severity of FMS symptoms and pain scores, as well as improved sleep quality and, in particular, improved depression symptoms, thus improving quality of life.

Ozone medical gas is a treatment method that provides treatment for many diseases with wide application areas and low side effect incidence. Ozone is a 3-molecule atom and is less stable than oxygen. Therefore, its biological response is higher than oxygen (*Özcan et al., 2019*). There is no consensus on a standard dose in the literature.

The pathophysiology of pain symptoms in FMS is explained by the presence of the central sensitization phenomenon, which is characterized by dysfunction of the neurocircuits that provide perception, transmission and processing of afferent nociceptive

Table 2 Spearmen correlation analysis between non-normally distributed data.

| | | Age | Visual Analog Scale-BT | Visual Analog Scale-AT | SF-36-Physical role-BT | SF-36-Physical role-AT | SF-36-Emotional role-BT | SF-36-Emotional role-AT | Sf-36-Social Functioning-BT | Sf-36-Social Functioning-AT | SF-36-Pain-BT | SF-36-Pain-AT | SF-36 General Health-BT | SF-36 General Health-AT |
|---|---|---|---|---|---|---|---|---|---|---|---|---|---|---|
| Age | r | 1,000 | 0.053 | 0.125 | −0.081 | −0.050 | −0.066 | −0.068 | −0.023 | −0.273* | 0.184 | −0.073 | 0.205 | −0.168 |
| | p | – | 0.661 | 0.303 | 0.504 | 0.682 | 0.589 | 0.573 | 0.849 | **0.022** | 0.128 | 0.546 | 0.089 | 0.165 |
| Visual Analog Scale-BT | r | 0.053 | 1,000 | **0.310**\*\* | −0.163 | −0.209 | −0.126 | −0.274\* | **−0.364**\*\* | −0.153 | **−0.489**\*\* | −0.248\* | **−0.356**\*\* | −0.447\*\* |
| | p | 0.661 | – | **0.009** | 0.172 | 0.082 | 0.290 | **0.022** | **0.002** | 0.206 | **0.000** | **0.039** | 0.002 | 0.000 |
| Visual Analog Scale-AT | r | 0.125 | **0.310**\*\* | 1,000 | −0.267\* | −0.499\*\* | −0.290\* | −0.452\*\* | −0.181 | −0.211 | −0.255\* | −0.658\*\* | −0.041 | −0.449\*\* |
| | p | 0.303 | **0.009** | – | **0.026** | **0.000** | **0.015** | **0.000** | 0.133 | 0.079 | **0.033** | **0.000** | 0.734 | 0.000 |
| SF-36-Physical role-BT | r | −0.066 | −0.163 | −0.267\* | 1,000 | **0.375**\*\* | **0.497**\*\* | **0.312**\*\* | **0.420**\*\* | **0.292**\* | **0.365**\*\* | **0.299**\* | 0.226 | 0.190 |
| | p | 0.589 | 0.172 | **0.026** | – | **0.001** | **0.000** | **0.009** | **0.000** | **0.014** | **0.002** | **0.012** | 0.057 | 0.116 |
| SF-36-Physical role-AT | r | −0.068 | −0.209 | −0.499\*\* | 0.375\*\* | 10.000 | 0.159 | **0.634**\*\* | **0.253**\* | **0.431**\*\* | 0.184 | **0.473**\*\* | 0.097 | 0.443\*\* |
| | p | 0.573 | 0.082 | **0.000** | **0.001** | – | 0.190 | **0.000** | **0.034** | **0.000** | 0.128 | **0.000** | 0.426 | 0.000 |
| SF-36-Emotional role-BT | r | −0.023 | −0.126 | −0.290\* | **0.497**\*\* | 0.159 | 1,000 | **0.348**\*\* | **0.427**\*\* | 0.067 | **0.381**\*\* | **0.251**\* | 0.086 | 0.011 |
| | p | 0.849 | 0.290 | **0.015** | **0.000** | 0.190 | – | **0.003** | **0.000** | 0.579 | **0.001** | **0.036** | 0.472 | 0.925 |
| SF-36-Emotional role-AT | r | −0.273\* | −0.274\* | −0.452\*\* | 0.312\*\* | 0.634\*\* | 0.348\*\* | 1,000 | **0.296**\* | **0.524**\*\* | 0.219 | **0.455**\*\* | 0.152 | 0.385\*\* |
| | p | **0.022** | **0.022** | **0.000** | **0.009** | **0.000** | **0.003** | – | **0.013** | **0.000** | 0.068 | **0.000** | 0.208 | 0.001 |
| Sf-36-Social Functioning-BT | r | 0.184 | −0.364\*\* | −0.181 | **0.420**\*\* | 0.253\* | **0.427**\*\* | 0.296\* | 1,000 | 0.425\*\* | **0.514**\*\* | **0.318**\*\* | **0.326**\*\* | **0.310**\*\* |
| | p | 0.128 | **0.002** | 0.133 | **0.000** | **0.034** | **0.000** | **0.013** | – | 0.000 | **0.000** | **0.007** | 0.005 | 0.009 |
| Sf-36-Social Functioning-AT | r | −0.073 | −0.153 | −0.211 | 0.292\* | 0.431\*\* | 0.067 | 0.524\*\* | 0.425\*\* | 1,000 | 0.108 | **0.372**\*\* | **0.356**\*\* | **0.405**\*\* |
| | p | 0.546 | 0.206 | 0.079 | **0.014** | **0.000** | 0.579 | **0.000** | **0.000** | – | 0.372 | **0.002** | 0.002 | 0.000 |
| SF-36-Pain-BT | r | 0.205 | **−0.489**\*\* | −0.255\* | 0.365\*\* | 0.184 | **0.381**\*\* | 0.219 | **0.514**\*\* | 0.108 | 1,000 | **0.338**\*\* | **0.356**\*\* | 0.278\* |
| | p | 0.089 | **0.000** | **0.033** | **0.002** | 0.128 | **0.001** | 0.068 | **0.000** | 0.372 | – | **0.004** | 0.002 | 0.020 |
| SF-36-Pain-AT | r | −0.168 | −0.248\* | −0.658\*\* | 0.299\* | 0.473\*\* | 0.251\* | 0.455\*\* | 0.318\*\* | 0.372\*\* | 0.338\*\* | 1,000 | 0.214 | 0.581\*\* |
| | p | 0.165 | **0.039** | **0.000** | **0.012** | **0.000** | **0.036** | **0.000** | **0.007** | **0.002** | **0.004** | – | 0.076 | 0.000 |
| SF-36 General Health-BT | r | −0.028 | −0.356\*\* | −0.041 | 0.226 | 0.097 | 0.086 | 0.152 | **0.326**\*\* | **0.356**\*\* | **0.356**\*\* | 0.214 | 1,000 | 0.504\*\* |
| | p | 0.818 | **0.002** | 0.734 | 0.057 | 0.426 | 0.472 | 0.208 | **0.005** | **0.002** | **0.002** | 0.076 | – | 0.000 |
| SF-36 General Health-AT | r | 0.054 | −0.447\*\* | −0.449\*\* | 0.190 | 0.443\*\* | 0.011 | 0.385\*\* | 0.310\*\* | 0.405\*\* | 0.278\* | 0.581\*\* | 0.504\*\* | 1,000 |
| | p | 0.659 | **0.000** | **0.000** | 0.116 | **0.000** | 0.925 | **0.001** | **0.009** | **0.000** | **0.020** | **0.000** | **0.000** | – |

Note:
BT: Before Treatment, AT: After Treatment, bold font: statistically significant, *: correlation is significant below 0.05, **: correlation is significant below 0.001.

stimuli (*Siracusa et al., 2021*). Nitric oxide causes the secretion of excitatory animosities and Substance P from the presynaptic afferent terminals. Thus, the dorsal horn becomes hyperexcitable (*Schaible, Ebersberger & von Banchet, 2001*).

There is also evidence of abnormalities in the oxygenation and/or oxidative systems in muscle metabolism and structure in patients with FMS. These structural damages may contribute to poor oxygen diffusion, decreased oxidative phosphorylation, and decreased ATP synthesis, which may further increase oxidative stress and

peroxidation of membrane lipids (*Schaible, Ebersberger & von Banchet, 2001*). Oxidative stress is even more pronounced in patients with FMS, where fatigue is more pronounced (*Chung et al., 2009*).

*Tirelli et al. (2022)* found that more than three-quarters of patients with FMS who applied MAH improved their arthralgic (pain and/or fatigue) symptoms following ozone therapy and described it as a promising treatment. In addition, it has been shown that when MAH and physiotherapy are applied together, it increases the effects of physiotherapy and is more effective in eliminating pain scores than when physiotherapy is applied alone (*Karakoyun, Yalkın & Yanartaş, 2023*). Similarly, in our study, we found significant decreases in pain scores after treatment. The reduction of pain in our patients is seen as another reason for improving the quality of life.

Another interesting issue is the effect of MAH on functional status, quality of life, depression and sleep. Ozone therapy *via* rectal insufflation in patients diagnosed with FMS seems to be beneficial mainly for the physical symptoms of fibromyalgia, although it also provides improvement in anxiety and depression (*Hidalgo-Tallón et al., 2013*). It has also been determined that MAH causes a decrease in sensitive points, oxidative stress level and FIQ score in patients diagnosed with FMS. Another effect is that MAH allows patients diagnosed with FMS to face life more vividly and with less medication use, thus reducing harmful side effects caused by medication (*Moreno-Fernández et al., 2019*). Apart from this, it has been reported that MAH contributes to the improvement of depression and provides improvement in quality of life and general health status (*Gazioğlu Türkyılmaz, Rumeli & Bakır, 2021*). After MAH application, the increase in serotonin levels and the decrease in oxidation products such as reactive oxygen species (ROPs) and lipid oxidation products (LOPs) provide improvements in sleep and pain pathways, positively affecting patient health in many parameters such as quality of life, sleep health and functional status (*Gazioğlu Türkyılmaz, Rumeli & Bakır, 2021*; *Moreno-Fernández et al., 2019*). It was surprising to see the beneficial effects observed with ozone therapy, showing clinically significant improvement in the severity of FMS symptoms and pain scores, as well as significantly improving sleep and quality of life.

Another important issue is the evaluation of the effectiveness of ozone application doses and the number of sessions in different studies. *Sucuoğlu & Soydaş (2023)* applied 10 sessions of MAH and 5 ml of minor autohemotherapy at the appropriate dosage in the FMS patient group: 15 g/ml for the first two sessions, 20 g/ml for the next four sessions, and 25 g/ml for the last four sessions. They observed significant improvements in sleep scores, but no significant decrease in FIQ scores was observed after treatment compared with the placebo group. *Gazioğlu Türkyılmaz, Rumeli & Bakır (2021)* used the same application method as our treatment and observed significant decreases in VAS, FIQ, and SF-36 scores. *Moreno-Fernández et al. (2019)* obtained 150 ml of blood and applied 30 μg/ml in the first three sessions, 40 μg/ml in the fourth session, 50 μg/ml in the fifth session, and 60 μg/ml in the final five sessions. Consequently, they observed significant decreases in FIQ scores and tender point counts. These results indicate that different results can be obtained with different application methods and that future studies could standardize the ozone application protocol. However, it should be kept in mind that each

patient should be evaluated individually, along with their clinical findings and comorbidities.

Another interesting aspect of our study was the evaluation of the effectiveness of MAH in the geriatric fibromyalgia group compared to the non-geriatric group. Sleep scores were significantly lower in the geriatric group before treatment. Similarly, sleep scores and SF-36 physical function scores were significantly lower after treatment. No difference was found in other parameters before and after treatment. FM is most commonly seen between the ages of 30 and 50, and its overall prevalence is reported to be between 2% and 4%, although the prevalence in elderly patients is unknown. A study in the literature reported a greater than 7% increase in the diagnosis rate in women aged 60 to 79 (*Minerbi & Fitzcharles, 2021*). Other possible pathologies should be excluded in the geriatric group to avoid unnecessary testing and overtreatment. Potential inflammatory rheumatological diseases (polymyalgia rheumatica), subclinical infectious conditions, nutritional disorders, endocrine disorders (electrolyte disorders and hypothyroidism), and malignancies should be considered in the differential diagnosis and excluded before making a diagnosis of fibromyalgia (*Häuser et al., 2017*). Sleep disturbances are a common problem in elderly patients with FM, and bruxism, sleep apnea, or gastroesophageal acid reflux disease should be considered in the differential diagnosis. Given that elderly patients are prone to complications from polypharmacy, it is important to consider the use of medical agents for the treatment of sleep disorders (*Hallegua & Wallace, 2009*). In our study, we found that MAH yielded similar results in elderly fibromyalgia patients as in non-elderly patients. However, we emphasize that more generalizable results may be obtained in future larger patient groups. Sleep disturbances are common in elderly patients with FM, and bruxism, sleep apnea, or gastroesophageal acid reflux disease should be considered in their differential diagnosis. Given that elderly patients are prone to complications from polypharmacy, it is important to consider carefully when using medical agents to treat sleep disorders (*Hallegua & Wallace, 2009*). In our study, we found that MAH yielded similar results in elderly fibromyalgia patients as in non-elderly patients. However, we emphasize that future studies with larger patient groups may yield more generalizable results.

This study has some limitations. High number of female patients, lack of long-term follow-up results and uncertainty of the period from the initial diagnosis to the present are some of these limitations. Because there is no standard methodology in studies on ozone therapy, the age parameters of the participants selected for the study span a wide range, and the results from the predominantly female participants are generalized to the general population. This led to a similar situation in our study. A further limitation is that no attempt is made to correct for confounding factors or to assess the potential effect of regression to the mean. Other limitations of our study include the fact that it was designed retrospectively and that it was not randomized controlled. Current results show that ozone therapy is effective in FMS. However, more randomized controlled studies are needed. It seems particularly important to investigate the optimum ozone dosage, number of sessions per week, and length of the treatment cycle required to achieve maximum therapeutic benefit.

We believe that future randomized controlled studies with large samples and long-term follow-up will clarify the effectiveness of treatment. We believe that samples containing markers of blood antioxidant levels that can be taken before and after treatment can also strengthen future studies. It is clear that future studies will be able to provide more robust and generalizable results about the effectiveness of ozone therapy by learning from the limitations we experienced in our study and taking our recommendations into consideration.

## CONCLUSION

This study found that MAH is an encouraging and successful treatment strategy, providing positive effects on pain, functional status, quality of life and sleep in patients diagnosed with fibromyalgia. Due to MAH's low side effects and promising results, it can be considered as a complementary and integrative treatment in the patient group diagnosed with fibromyalgia who do not receive adequate response from other treatments.

### Funding
The authors received no funding for this work.

### Competing Interests
The authors declare that they have no competing interests.

### Author Contributions
- Oktay Faysal Tertemiz conceived and designed the experiments, performed the experiments, analyzed the data, prepared figures and/or tables, authored or reviewed drafts of the article, and approved the final draft.
- Erkan Ozduran conceived and designed the experiments, performed the experiments, analyzed the data, prepared figures and/or tables, authored or reviewed drafts of the article, and approved the final draft.
- Sinan Gursoy conceived and designed the experiments, performed the experiments, analyzed the data, prepared figures and/or tables, authored or reviewed drafts of the article, and approved the final draft.
- Volkan Hanci conceived and designed the experiments, performed the experiments, analyzed the data, prepared figures and/or tables, authored or reviewed drafts of the article, and approved the final draft.
- Kenan Kaygusuz conceived and designed the experiments, performed the experiments, analyzed the data, prepared figures and/or tables, authored or reviewed drafts of the article, and approved the final draft.
- Ilhan Celil Ozbek conceived and designed the experiments, performed the experiments, analyzed the data, prepared figures and/or tables, authored or reviewed drafts of the article, and approved the final draft.

- Mustafa Kurçaloğlu conceived and designed the experiments, performed the experiments, analyzed the data, prepared figures and/or tables, authored or reviewed drafts of the article, and approved the final draft.
- Ülkü Sabuncu conceived and designed the experiments, performed the experiments, analyzed the data, prepared figures and/or tables, authored or reviewed drafts of the article, and approved the final draft.

### Ethics

The following information was supplied relating to ethical approvals (*i.e.*, approving body and any reference numbers):

This research was approved by Sivas Cumhuriyet University Non-Interventional Ethics Committee/Turkey (2023/09-11 GOA)

### Data Availability

The raw data is available in the Supplemental Files.

### Supplemental Information

Supplemental information for this article can be found online at http://dx.doi.org/10.7717/peerj.20475#supplemental-information.

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
