# Peer review of "Effect of major ozone autohemotherapy in fibromyalgia syndrome: a retrospective study"

_PeerJ, doi:10.7717/peerj.20475_

## Round 0.1 · original submission · Major Revisions

· Academic Editor

Major Revisions

**Language Note:** When preparing your next revision, please ensure that your manuscript is reviewed either by a colleague who is proficient in English and familiar with the subject matter, or by a professional editing service. PeerJ offers language editing services; if you are interested, you may contact us at [email protected] for pricing details. Kindly include your manuscript number and title in your inquiry. – PeerJ Staff

Reviewer 1 ·

Basic reporting

• The manuscript is generally well structured and written in clear, professional English. However, the reporting of methodological aspects and literature use presents significant limitations.

• The number of Turkish-language references is notably high (11 out of 23), which limits the international relevance of the discussion. Moreover, 3 of the references cite the authors' own previous work. While this is understandable to some extent, it raises concerns about balance and neutrality. The authors are encouraged to incorporate more high-quality, up-to-date international sources.

• The citation of Sucuoğlu & Soydaş (2023) as supportive of the effectiveness of MAH is misleading. This study is a high-quality RCT whose primary outcome (FIQ total score) showed no superiority of ozone therapy over placebo. The selective citation of only secondary, favorable outcomes without acknowledging the primary null result misrepresents the literature.

Experimental design

• The study’s design—a retrospective, single-arm, uncontrolled observational study—limits its ability to support causal claims. Improvements in post-treatment measures could be due to placebo effects, regression to the mean, or the natural course of fibromyalgia, rather than a true pharmacological effect of MAH.

• The inclusion criterion of “at least 3 years of unresponsiveness to medical treatment” is inadequately defined. The manuscript does not specify which treatments were attempted, in what doses or durations, or what metrics were used to determine treatment failure.

• The average duration of fibromyalgia diagnosis among patients is not reported. Descriptive statistics (mean, SD, min–max) on this variable would enhance transparency.

• The statement that MAH was applied “in accordance with the Madrid Declaration on Ozone Therapy” lacks sufficient detail. The actual ozone dose, concentration (µg/mL), and total amount per session (e.g., gamma) are not provided. Moreover, it is unclear whether the dosage remained constant across sessions or was titrated, which is an important aspect of treatment replicability.

• A power analysis is mentioned, but it is inappropriate for a single-arm, retrospective study design and gives a misleading impression of statistical rigor.

Validity of the findings

• The lack of a control group severely limits the internal validity of the study.

• No attempts were made to correct for confounders or to assess the potential impact of regression to the mean.

• Although the authors report statistically significant changes in outcome measures, causality cannot be established with the current design.

• The presence of 12 patients over the age of 65 (approximately 17% of the sample) raises additional methodological concerns. Fibromyalgia diagnosis in elderly populations is complex and controversial, as symptoms may overlap with other age-related conditions such as osteoarthritis or polymyalgia rheumatica. The study does not report how differential diagnoses were excluded or whether patients underwent specialist evaluations (e.g., rheumatology, neurology, geriatrics).

• The authors should perform subgroup analyses by age group and discuss the diagnostic challenges in elderly populations explicitly in the discussion or limitations section.

Additional comments

• While the topic is relevant and the treatment approach is novel, the current version of the manuscript contains several critical flaws in methodology, data transparency, and interpretation.

• The discussion section should be expanded to address unexpected findings, possible biases (e.g., selection bias), and limitations in greater depth.

• Cost-effectiveness analysis, or at least a discussion on it, would strengthen the practical applicability of the results.

Reviewer 2 ·

Basic reporting

The work is interesting and touches on a research area with significant clinical and social impact. The research objective is well-defined and consistent with the stated premises. The manuscript is overall written in professional and comprehensible English, although some passages require careful linguistic revision to improve clarity and fluency (for example, in the sentence on line 80, "It is also to investigate...", the grammatical structure appears unnatural).

The introduction appears brief and lacking in detail. In particular, the description of the biochemical mechanisms through which ozone exerts its therapeutic action is insufficiently detailed. In my opinion, it would be important for the authors to clarify in more detail the biochemical mechanisms of ozone action and why these may be relevant for fibromyalgia patients. Currently, some key assertions are supported by a few studies and are poorly articulated. For example, the sentence on line 74 would require more bibliographic support to be convincing (only one study is cited).

I suggest enriching this section with recent and robust bibliographic references to better clarify the scientific context of ozone therapy in fibromyalgia. Relevant topics to explore include: the short- and medium-term clinical efficacy of ozone autohemotherapy in managing fibromyalgia symptoms such as pain, fatigue, and sleep disturbances; the role of oxidative stress modulation and immune system regulation in chronic pain conditions; and the broader therapeutic applications of ozone therapy in musculoskeletal medicine, including its anti-inflammatory and analgesic mechanisms, safety considerations, and administration protocols.

Regarding the overall structure of the manuscript, it is well organized and follows the standard sections of a scientific article (Introduction, Methods, Results, Discussion, Conclusions). Figures and tables are relevant and correctly labeled.

Experimental design

This work presents itself as original research that clearly falls within the journal's objectives and focus. The research question is well-defined, relevant, and clearly aimed at filling a significant gap in knowledge in the management of fibromyalgia. The methodology appears rigorous. The authors provide a detailed explanation of the limitations of the observational study, so their effort to discuss these limitations is commendable. Perhaps it would be appropriate to expand on future research prospects and the clinical implications of these findings (such as implementing new patient care protocols, etc.). The tests used are well-known and validated in the literature, allowing for sufficient rigor in measuring the variables.

Validity of the findings

The results are presented clearly and consistently with the study's objectives. The tables and figures are well-designed, easy to read, and effectively support understanding of the data. The conclusions are well-formulated, directly related to the research question, and based on the results.

Reviewer 3 ·

Basic reporting

After thoroughly reading the article, I found that the English language was generally well used throughout the text. The topic is contemporary, novel, and of general interest to the scientific community.

Experimental design

• Overall, the study is well-designed and addresses a clinically relevant topic.

• However, it would benefit from the inclusion of more up-to-date references, especially from 2025 and beyond, to better contextualize the findings within the current literature.

Validity of the findings

-

Additional comments

Abstract
• The Methods section of the abstract needs to be completely restructured. The ozone dosage and administration protocol must be clearly defined.
• The inclusion of the ethics committee information in the abstract is unnecessary and should be removed.
• The Conclusion is overly simplistic and lacks sufficient emphasis on the study’s findings. The results should be more explicitly and quantitatively presented.

Introduction
• The introduction should include a brief overview of the current treatment modalities for fibromyalgia syndrome (FMS), including both pharmacological and non-pharmacological options.
• The rationale and mechanism of action of ozone therapy, particularly in the context of FMS, should be summarized.
• It is essential to clearly define the aim of the study and articulate the knowledge gap that this study seeks to address.
• Line 78: The statement describing the study’s objective is vague and inadequate. Please revise to explicitly state the study aim, followed by a clear hypothesis.

Methods
• The exclusion criteria for patient selection are insufficiently explained and should be elaborated.
• Clarify why the same author conducted the entire treatment process. This may introduce potential bias and should be addressed or justified.
• It would be helpful to briefly describe the clinical scoring systems used:
o Fibromyalgia Impact Questionnaire (FIQ) – functional capacity,
o Pittsburgh Sleep Quality Index (PSQI) – sleep quality,
o Short Form-36 (SF-36) – quality of life.
• The ozone dosage, concentration, and method of administration must be clearly detailed.
• I strongly recommend reviewing studies that compare different ozone administration protocols in musculoskeletal disorders
• It is also necessary to specify which concurrent treatments the patients received during the study period (e.g., NSAIDs, antidepressants, physiotherapy, psychotherapy, etc.).

Limitations
• The authors mention the small sample size as a limitation, but no power analysis is provided. If a power analysis was conducted and the sample size is adequate for the detected effect size, this should be clarified. Otherwise, this limitation needs to be supported statistically.

---

## Round 0.2 · Minor Revisions

· Academic Editor

Minor Revisions

Reviewer 1 ·

Basic reporting

No further comments. The manuscript is now clearly written, well-structured, and supported with sufficient and balanced international literature references.

Experimental design

No further comments. Methods have been adequately clarified, and sufficient detail is now provided to ensure reproducibility.

Validity of the findings

No further comments. The authors have clearly acknowledged the study’s limitations and strengthened the discussion of internal validity. Conclusions are now well aligned with the presented data.

Additional comments

I am satisfied with the revisions. The manuscript has been substantially improved and is suitable for acceptance in its current form.

Reviewer 2 ·

Basic reporting

The manuscript uses clear and professional English throughout. The terminology is appropriate for the scientific context.
The text is sufficiently detailed and includes extensive bibliographical references.
The article follows a professional and standard structure. The sections are clear and well-written. They explain the concepts better than the previous version, which has significantly improved.

Experimental design

The manuscript represents an original primary research that clearly falls within the journal's objectives.
The research question is well-defined, clinically relevant, and meaningful, and concerns the efficacy of major autohemotherapy in patients with fibromyalgia resistant to conventional treatments. The authors explicitly identify a gap in the current literature.
The research was conducted according to high technical and ethical standards, with ethics committee approval, written informed consent, and adherence to the Declaration of Helsinki. The methods are described in sufficient detail.

Validity of the findings

The rationale for replication and its benefits to the literature are sufficiently detailed.
All data are provided and robust, statistically valid, and adequately controlled.
The conclusions are clear, directly related to the original research question, and remain within the limitations of the supporting findings.

The limitations section is highly appreciated. It is transparent, detailed, and strengthens the study's credibility.
Overall, this version of the article is more compelling and clearer than the previous one.

Reviewer 3 ·

Basic reporting

This manuscript is an important study that highlights the significance of regenerative medicine in patients with fibromyalgia, and I consider it valuable for publication.

Experimental design

The study design is quite good; however, there are many articles comparing ozone therapy with dry needling and steroid injections, and evaluating its effects. I recommend citing more studies from 2024–2025.

Validity of the findings

-

Additional comments

More studies from the current literature on the applications of ozone should be reviewed and incorporated into the discussion. The study design is quite clear, and the data are meaningful.

---

## Round 0.3 · accepted · Accept

· Academic Editor

Accept

Thank you for carefully addressing all the reviewer comments and for updating the manuscript to include recently published citations. I am pleased to confirm that your article is now accepted for publication. Congratulations on your excellent and diligent work!

Reviewer 3 ·

Basic reporting

I would like to thank the authors for carefully addressing my previous comments and implementing the suggested revisions.

Experimental design

I would like to thank the authors for carefully addressing my previous comments and implementing the suggested revisions.

Validity of the findings

I would like to thank the authors for carefully addressing my previous comments and implementing the suggested revisions.

Additional comments

Acceptabl